# The Effect of Different Writing Tasks on Linguistic Style: A Case Study of the ROC Story Cloze Task

## Abstract

A writer's style depends not just on personal traits but also on her intent and mental state. In this paper, we show how variants of the same writing task can lead to measurable differences in writing style. We present a case study based on the *story cloze task* (Mostafazadeh et al., 2016a), where annotators were assigned similar writing tasks with different constraints: (1) writing an entire story, (2) adding a story ending for a given story context, and (3) adding an incoherent ending to a story. We show that a simple linear classifier informed with stylistic features is able to successfully distinguish among the three cases, without even looking at the story context. In addition, our style-based classifier establishes a new state-of-the-art result on the story cloze challenge, substantially higher than previous results based on deep learning models. Our results demonstrate that different task framings can dramatically affect the way people write.

| Type | Example |
|------|---------|
| Original story | My mother loves clocks that chime. Her house is full of them. She sets them each a little different so she can hear them chime. It sounds like a bell tower during a wedding in her house all day. When I visit I stop them or I'd never be able to sleep at night. |
| Coherent story | Kathy went shopping. She found a pair of great shoes. The shoes were $300. She bought the shoes. She felt buyer's remorse after the purchase. |
| Incoherent story | Kathy went shopping. She found a pair of great shoes. The shoes were $300. She bought the shoes. Kathy hated buying shoes. |

Table 1: Examples of stories from the story cloze task (Mostafazadeh et al., 2016a). The first row shows an original story written by one author. The second and third rows show revised stories with two contrastive endings: a coherent ending and a incoherent one.

## 1 Introduction

Writing style is expressed through a range of linguistic elements such as words, sentence structure, and rhetorical devices. It is influenced by personal factors such as age and gender (Schler et al., 2006), by personality traits such as agreeableness and openness (Ireland and Mehl, 2014), as well as by mental states such as sentiment (Davidov et al., 2010), sarcasm (Tsur et al., 2010), and deception (Feng et al., 2012). In this paper, we study the extent to which writing style is affected by the nature of the writing task the writer was asked to perform, since different tasks likely engage different cognitive processes (Campbell and Pennebaker, 2003;

Banerjee et al., 2014).

We show that similar writing tasks with different constraints on the author can lead to measurable differences in her writing style. As a case study, we present experiments based on the recently introduced ROC story cloze task (Mostafazadeh et al., 2016a). In this task, authors were asked to write five-sentence self-contained stories, henceforth *original* stories. Then, each original story was given to a different author, who was shown only the first four sentences as a story context, and asked to write two contrasting story endings: a *right* (coherent) ending, and a *wrong*

(incoherent) ending. Framed as a story cloze task, the goal of this dataset is to serve as a commonsense challenge for NLP and AI research. Table 1 shows an example of an original story, a coherent story, and an incoherent story.

While the story cloze task was originally designed to be a story understanding challenge, its annotation process introduced three variants of the same writing task: writing an *original*, *right*, or *wrong* ending to a short story. In this paper, we show that a linear classifier informed by stylistic features can distinguish among the different endings to a large degree, even without looking at the story context (64.5–75.6% binary classification results).

Our results allow us to make a few key observations. First, people adopt a different writing style when asked to write coherent vs. incoherent story endings. Second, people change their writing style when writing the entire story on their own compared to writing only the final sentence for a given story context written by someone else.

In order to further validate our method, we also directly tackle the story cloze task. Adapting our classifier to the task, we obtain 72.4% accuracy, a 12.5% increase over the previously reported state-of-the-art (Salle et al., 2016). We also show that the style differences captured by our model can be combined with neural language models to make a better use of the story context. Our final model that combines context with stylistic features achieves 75.2%—an additional 2.8% gain, 15.3% better than the best published result.

The contributions of our study are threefold. First, findings from our study can potentially shed light on how different kinds of cognitive load influence the style of written language. Second, our results indicate that when designing new NLP tasks, special attention needs to be payed to the instructions given to authors. Third, we establish a new state-of-the-art result on the commonsense story cloze challenge.

## 2 Background: The Story Cloze Task

To understand how different writing tasks affect writing style, we focus on the *story cloze task* (Mostafazadeh et al., 2016a). While this task was developed to facilitate representation and learning of commonsense story understanding, its design included a few key choices which make it ideal for our study. We describe the task below.

**ROC stories.** The ROC story corpus consists of 49,255 five-sentence commonsense stories, collected on Amazon Mechanical Turk (AMT).[1] Workers were instructed to write a coherent self-contained story, which has a clear beginning and end. To collect a broad spectrum of commonsense knowledge, there was no imposed subject for the stories, which resulted in a wide range of different topics.

**Story cloze task.** After compiling the story corpus, the *story cloze task*—a task based on the corpus—was introduced. A subset of the stories was selected, and only the first four sentences of each story were presented to AMT workers. Workers were asked to write a pair of new story endings for each story context: one *right* and one *wrong*. Both endings are required to complete the story using one of the characters in the story context. Additionally, the ending is required to be "realistic and sensible" (Mostafazadeh et al., 2016a) when read out of context.

The resulting stories, both *right* and *wrong*, were then individually rated for coherence and meaningfulness by additional AMT workers. Only stories rated as simultaneously coherent with a *right* ending and neutral with a *wrong* ending were selected for the task. It is worth noting that workers rated the stories as a whole, not only the endings.

Based on the new stories, Mostafazadeh et al. (2016a) proposed the *story cloze task*. The task is simple: given a pair of stories that differ only in their endings, the system decides which ending is *right* and which is *wrong*. The official training data contains only the original stories (without alternative endings), while development and test data consist of the revised stories with alternative endings (for a different set of original stories that are not included in the training set). The task was suggested as an extensive evaluation framework: as a commonsense story understanding task, as the shared task for the Linking Models of Lexical, Sentential and Discourse-level Semantics workshop (LSDSem 2017), and as a testbed for vector-space evaluation (Mostafazadeh et al., 2016b).

Interestingly, at the time of this submission, 10 months after the task was first introduced, the published benchmark on this task is still below

---

[1]Recently, an additional 53K stories were released, which results in roughly 100K stories.

60% (Salle et al., 2016).[2] This comes in contrast to other recent similar machine reading tasks such as CNN/DailyMail (Hermann et al., 2015), SNLI (Bowman et al., 2015), LAMBADA (Paperno et al., 2016) and SQuAD (Rajpurkar et al., 2016), for which results improved dramatically over a similar or shorter period of time. This suggests that this task is challenging and that high performance is hard to achieve.

In addition, Mostafazadeh et al. (2016a) made substantial efforts to ensure the quality of this dataset. First, each pair of endings was written by the same author, which ensured that style differences between authors could not be used to solve the task. Furthermore, Mostafazadeh et al. implemented nine baselines for the task, using surface level features as well as narrative-informed ones, and showed that each of them reached roughly chance-level. These results suggest that real understanding of text is required in order to solve the task.

**Different writing tasks in the story cloze task.** Several key design decisions make the task an interesting testbed for the purpose of this study. First, the training set for the task (ROC Stories corpus) is not a training sample in the usual sense,[3] as it contains only positive (*right*) samples, and not negative (*wrong*) ones.

On top of that, the *original* endings, which serve as positive training samples, were generated differently from the *right* samples, which serve as the positive samples in the development and test sets. While the former are part of a single coherent story written by the same author, the latter were generated by letting an author read four sentences, and then asking her to generate a fifth *right* ending.

Finally, although the *right* and *wrong* sentences were generated by the same author, the tasks for generating them were quite different: in one case, the author was asked to write a *right* ending, which would create a coherent five-sentence story along with the other four sentences. In the other case, the author was asked to write a *wrong* ending, which would result in an incoherent five-sentence story.

---

[2]The LSDSem 2017 shared task website (https://competitions.codalab.org/competitions/15333) does report higher results, which are still unpublished along with the underlying methodology.

[3]I.e., the training instances are not drawn from a population similar to the one that future testing instances will be drawn from.

## 3 Surface Analysis of the Story Cloze Task

We begin by computing several characteristics of the three types of endings: *original* endings (from the ROC story corpus training set), *right* endings and *wrong* endings (both from the story cloze task development set). Our analysis reveals several style differences between different groups. First, *original* endings are on average longer (11 words per sentence) than *right* endings (8.75 words), which are in turn slightly longer than *wrong* ones (8.47 words). Previous work has shown that sentence length is also indicative of whether a text was deceptive (Yancheva and Rudzicz, 2013; Qin et al., 2004). Although writing *wrong* sentences is not the same as lying, it is not entirely surprising to observe similar trends in both tasks.

Second, Figure 1a shows the distribution of five frequent POS tags in all three groups. The figure shows that both *original* and *right* endings use pronouns more frequently than *wrong* endings. Once again, deceptive text is also characterized by fewer pronouns compared to truthful text (Newman et al., 2003).

Finally, Figure 1b presents the distribution of five frequent words across the different groups. The figure shows that *original* endings use coordinations ("and") more than *right* endings, and substantially more than *wrong* ones. Furthermore, *original* and *right* endings seem to prefer enthusiastic language (e.g., "!"), while *wrong* endings tend to use more negative language ("hates"), similarly as deceptive text (Newman et al., 2003). Next we show that these style differences are not anecdotal, but can be used to distinguish among the different types of story endings.

## 4 Model

The goal of this paper is to determine the extent to which different writing constraints lead the authors to adopt different writing styles. In order to answer these questions, we use simple methods that have been shown to be very effective for recognizing style (see Section 9). We describe our model below.

We train a logistic regression classifier to categorize an ending, either as *right* vs. *wrong* or as *original* vs. new (*right*). Each feature vector is computed using the words in one ending, without considering earlier parts of the story. We use the following style features.

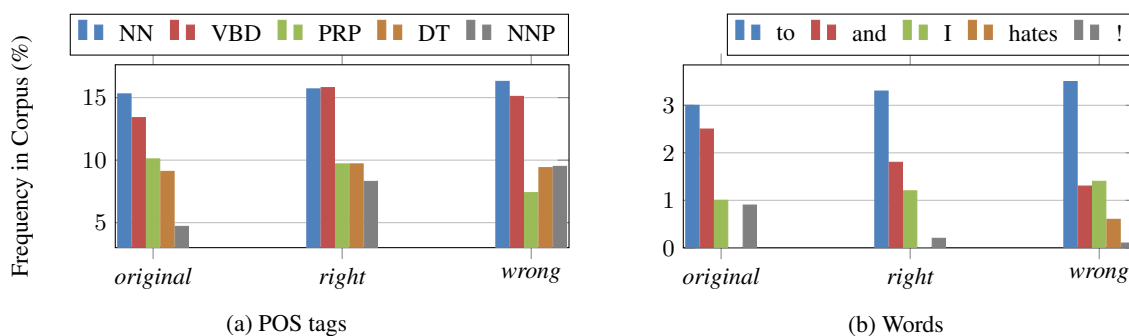

(a) POS tags (b) Words

Figure 1: The distribution of five frequent POS tags (1a) and words (1b) across *original* endings (story cloze training set), and *right* and *wrong* endings (from the story cloze task).

- *Length*. The number of words in the sentence.

- *Word n-grams*. We use sequences of 1-5 words. Following Tsur et al. (2010) and Schwartz et al. (2013b), we distinguish between high frequency and low frequency words. Specifically, we replace content words (nouns, verbs, adjectives, and adverbs), which are often low frequency, with their part-of-speech tags.

- *Character n-grams*. Character $n$-grams are one of the most useful features in identifying author style (Stamatatos, 2009). We use character 4-grams.

## 5 Experiments

We design two experiments to answer our research questions. The first is an attempt to distinguish between *right* and *wrong* endings, the second between *original* endings and new (*right*) endings. We describe both experiments below.

**Experiment 1: right/wrong endings.** The goal of this experiment is to measure the extent to which style features capture differences between the *right* and *wrong* endings. As the story cloze task doesn't have a training corpus for the *right* and *wrong* endings (see Section 2), we use the development set as our training set, holding out 10% for development (3,366 training endings, 374 for development). We keep the story cloze test set as is (3,742 endings).

It is worth noting that our classification task is slightly different from the story cloze task. Instead of classifying pairs of endings, one which is *right* and another which is *wrong*, our classifier decides about each ending individually, whether it is *right* (positive instance) or *wrong* (negative instance). By ignoring the coupling between *right* and *wrong* pairs, we are able to decrease the impact of author-specific style differences, and focus on the difference between the styles accompanied with *right* and *wrong* writings.

**Experiment 2: original/new endings.** Here the goal is to measure whether writing the ending as part of a story imposes different style compared to writing a new (*right*) ending to an existing story. We use the endings of the ROC stories as our *original* samples and *right* endings from the story cloze task as *new* samples. As there are far more *original* instances than *new* instances, we randomly select the same number of *original* instances as we have *new* instances (3,366 training endings, 374 development endings, and 3,742 test endings). We randomly sample 5 *original* sets and repeat the classification experiments. We report the average classification result.

**Experimental setup.** In both experiments, we add a START symbol at the beginning of each sentence.[4] For computing our features, we keep $n$-gram (character or word) features that occur at least five times in the training set. All feature values are normalized to $[0, 1]$. For the POS features, we tag all endings with the Spacy POS tagger.[5] We use Python's sklearn logistic regression implementation (Pedregosa et al., 2011) with $L_2$ regularization, performing grid search on the development set to tune a single hyperparameter—the regularization parameter.

---

[4]Virtually all sentences end with a period or an exclamation mark, so we do not add a STOP symbol.
[5]http://spacy.io/

| Experiment | Accuracy |
|---|---|
| *right vs. wrong* | 0.645 |
| *original vs. right* | 0.685 |
| *original vs. wrong* | 0.756 |

Table 2: Results of experiments 1 (*right vs. wrong*) and 2 (*original vs. right (new)*). The bottom row shows an additional experiment which classifies *original vs. wrong (new)* endings. In all cases, our setup implies a 50% random baseline.

## 6 Results

Table 2 shows our results. In both experiments, our model achieves performance well above what would be expected under chance (50% by design). Noting again that our model ignores the story context (the preceding four sentences), our model is unable to capture any notion of coherence. This finding provides strong evidence that the authors' style was affected by the writing task they were given to perform. We further applied the model to a third task, *original* vs. *wrong (new)* endings, and even stronger differences were apparent to the model, allowing over 75% accuracy.

**Story cloze task.** The results of Experiment 1 indicate that *right* and *wrong* endings are characterized by different styles. In order to further estimate the quality of our classification results, we tackle the story cloze task using our classifier. This classification task is more constrained than Experiment 1, as two endings are given and the question is which is *right* and which is *wrong*. We apply the classifier from Experiment 1 as follows: if it assigns different labels to the two given endings, we keep them. Otherwise, the label whose posterior probability is lower is reversed.

Table 3 shows our results on the story cloze test set. Our classifier obtains 72.4% accuracy, 12.5% (absolute) higher than the published state-of-the-art result on the task (Salle et al., 2016). Importantly, unlike previous approaches, our classifier does not require the story corpus training data, and in fact doesn't even consider the first four sentences of the story in question. These numbers further support the claim that the styles of *right* and *wrong* endings are indeed very different.

**Combination with a neural language model.** We investigate whether our model can benefit from state-of-the-art text comprehension models,

| Model | Acc. |
|---|---|
| †DSSM (Mostafazadeh et al., 2016a) | 0.585 |
| †LexVec (Salle et al., 2016) | 0.599 |
| †RNN | 0.677 |
| **Ours** | **0.724** |
| †**Combined (ours + RNN)** | **0.752** |
| †Human judgment | 1.000 |

Table 3: Results on the test set of the story cloze task. The upper block shows published results, the middle block are our results. LexVec results are taken from (Speer et al., 2016). Human judgement scores are taken from (Mostafazadeh et al., 2016a). Methods marked with (†) use the story context in order to make a prediction.

for which this task was designed. Specifically, we experiment with an LSTM-based (Hochreiter and Schmidhuber, 1997) recurrent neural network language model (RNNLM; Mikolov et al., 2010). Unlike the model in this paper, which only considers the story endings, this language model follows the protocol suggested by the story cloze task designers, and harnesses their ROC Stories training set, which consists of single-ending stories, as well as the story context for each pair of endings. We show that adding our features to this powerful language model gives improvements over our classifier as well as the language model.

We train the RNNLM using a single-layer LSTM of hidden dimension 512. We use the ROC stories for training,[6] setting aside 10% for validation of the language model. We replace all words occurring less than 3 times with a special out-of-vocabulary character, yielding a vocabulary size of 21,582. Only during training, we apply a dropout rate of 60% while running the LSTM over all 5 sentences of the stories. Using the Adam optimizer (Kingma and Ba, 2015) and a learning rate of $\eta = .001$, we train to minimize cross-entropy.

To apply the language model to the classification problem, we select as *right* the ending with the higher value of

$$\frac{p_\theta(\text{ending} \mid \text{story})}{p_\theta(\text{ending})} \quad (1)$$

The intuition is that a *right* ending should be unsurprising (to the model) given the four preceding

---

[6]We use the extended, 100K stories corpus (see Section 2).

| Feature Type | Accuracy |
|---|---|
| Word $n$-grams | 0.612 |
| Character $n$-grams | 0.639 |
| Full model | 0.645 |

Table 4: Results on Experiment 1 with different subsets of features.

sentences of the story (the numerator), controlling for the inherent surprisingness of the words in that ending (the denominator).

On its own, our neural language model performs moderately well on the story cloze test. Selecting endings based on $p_\theta(\text{ending} \mid \text{story})$ (i.e., the numerator of Equation 1), we obtained only 55% accuracy. The ratio in Equation 1 achieves 67.7% (see Table 3).[7]

We combine our linear model with the RNNLM by adding three features to our classifier: the numerator, denominator, and ratio in Equation 1, all in log space. We retrain our linear model with the new feature set, and gain 2.8% absolute, reaching 75.2% (15.3% better than the published state-of-the-art result). These results indicate that context-ignorant style features can be used to obtain high accuracy on the task, adding value even when context and a large training dataset are used.

## 7 Further Analysis

### 7.1 Most Discriminative Feature Types

A natural question that follows this study is which style features are most helpful in detecting the underlying task an author was asked to perform. To answer this question, we re-ran Experiment 1 with different sub-groups of features. Table 4 shows our results. Results show that character $n$-grams are the most effective style predictors, reaching within 0.6% of the full model, but that word $n$-grams also capture much of the signal, yielding 61.2%, which is only 3.3% worse than the full model. These findings are in line with previous work that used character $n$-grams along with other types of features to predict writing style (Schwartz et al., 2013b).

### 7.2 Most Salient Features

A follow-up question is which individual features contribute most to the classification process, as

________

[7]Note that taking the logarithm of the expression in Equation 1 gives the pointwise mutual information between the story and the ending, under the language model.

| *Right* | Freq. | *Wrong* | Freq. |
|---|---|---|---|
| 'ed .' | 6.5% | START NNP | 54.8% |
| 'and ' | 13.6% | NN . | 47.5% |
| JJ | 45.8% | NN NN . | 5.1% |
| to VB | 20.1% | VBG | 10.1% |
| 'd th' | 10.9% | START NNP VBD | 41.9% |

Table 5: The top 5 most heavily weighted features for predicting *right* vs. *wrong* endings, along with their frequency in our story cloze training set.

these could shed light on the stylistic differences imposed by each of the writing tasks.

In order to answer this question, we consider the highest absolute positive and negative coefficients in the logistic regression classifier in both Experiments 1 and 2, an approach widely used as a method of extracting the most salient features (Nguyen et al., 2013; Burke et al., 2013; Brooks et al., 2013). It is worth noting that its reliability is not entirely clear, since linear models like logistic regression can assign large coefficients to rare features (Yano et al., 2012). To mitigate this concern, we consider only features appearing in at least 5% of the endings in our training set.

**Experiment 1.** Table 5 shows the most salient features for *right* (coherent) and *wrong* (incoherent) endings in Experiment 1, along with their corpus frequency. The table shows a few interesting trends. First, authors tend to structure their sentences differently when writing coherent vs. incoherent endings. For instance, incoherent endings are more likely to start with a proper noun and end with a common noun, while coherent endings have a greater tendency to end with a past tense verb.

Second, *right* endings will make wider use of coordination structures, as well as adjectives. The latter might indicate that writing coherent stories inspires the authors to write more descriptive text compared to incoherent ones, as is the case in truthful vs. deceptive text (Ott et al., 2011). Finally, we notice a few syntactic differences: *right* endings will more often use infinite verb structure, while *wrong* endings prefer gerunds (VBG).

**Experiment 2.** Table 6 shows the same analysis for Experiment 2. As noted in Section 2, *original* endings tend to be much longer, which is indeed the most salient feature for them. An interesting observation is that exclamation marks are a strong indication for an *original* ending. This suggests

| *Original* | Freq. | *New* | Freq. |
|---|---|---|---|
| *length* | 100.0% | '.' | 93.0% |
| '!' | 6.1% | START NNP | 39.2% |
| NN | 78.9% | START NNP VBD | 29.0% |
| RB | 44.7% | NN . | 42.3% |
| ',' | 12.7% | the NN . | 10.6% |

Table 6: The top 5 most heavily weighted features for predicting *original* vs. new (*right*) endings. *length* is the sentence length feature (see Section 4).

that authors are more likely to show or evoke enthusiasm when writing their own text compared to ending an existing task.

Finally, when comparing the two groups of salient features from both experiments, we find an interesting trend. Several features, such as "START NNP" and "NN .", which indicate *wrong* sentences in Experiment 1, are used to predict new (i.e., *right*) endings in Experiment 2. This indicates that, for instance, incoherent endings have a stronger tendency to begin with a proper noun compared to coherent endings, which in turn are more likely to do so than original endings. This partially explains why distinguishing between *original* and *wrong* endings is an easier task compared to the other pairs (Section 6).

## 8 Discussion

**The effect of writing tasks on mental states.** In this paper we have shown that giving a writer different writing tasks affects her writing style in easily detected ways. Our results indicate that when authors are asked to write the last sentence of a five-sentence story, they will use different style to write a *right* ending compared to a *wrong* ending. We have also shown that writing the ending as part of one's own five-sentence story is very different than reading four sentences and then writing the fifth. Our findings hint that the nature of the writing task imposes a different mental state on the author, which is expressed in ways that can be observed using extremely simple automatic tools.

Previous work has shown that a writing task can affect mental state. For instance, writing deceptive text leads to a significant cognitive burden accompanied by a writing style that is different from truthful text (Newman et al., 2003; Banerjee et al., 2014). Writing tasks can even have a long-term effect, as writing emotional texts was observed to benefit both physical and mental health (Lepore and Smyth, 2002; Frattaroli, 2006). Campbell and Pennebaker (2003) also showed that the health benefits of writing emotional text are accompanied by changes in writing style, mostly in the use of pronouns.

Another line of work has shown that writing style is affected by mental state. First, an author's personality traits (e.g., depression, neuroticism, narcissism) affect her writing style (Schwartz et al., 2013a; Ireland and Mehl, 2014). Second, temporary changes, such as a romantic relationship (Ireland et al., 2011; Bowen et al., 2016), work collaboration (Tausczik, 2009; Gonzales et al., 2009), or negotiation (Ireland and Henderson, 2014) may also affect writing style. Finally, writing style can also change from one sentence to another, for instance between positive and negative text (Davidov et al., 2010) or when writing sarcastic text (Tsur et al., 2010).

This large body of work indicates a tight connection between writing tasks, mental states, and variation in writing style. This connection hints that the link discovered in this paper, between different writing tasks and resulting variation in writing style, involves differences in mental state. Further investigation is required in order to further validate this hypothesis.

**Design of NLP tasks.** Our study also provides important insights for the future design of NLP tasks. The story cloze task was very carefully designed. Many factors, such as topic diversity and temporal and causal relation diversity, were controlled for (Mostafazadeh et al., 2016a). The authors also made sure each pair of endings was written by the same author, partly in order to avoid author-specific style effects. Nonetheless, despite these efforts, several significant style differences can be found between the training and the test set, as well as between the positive and negative labels.

Our findings suggest that careful attention must be paid to instructions given to authors, especially in unnatural tasks such as writing a *wrong* ending. One way to avoid such problems is by using shorter text spans, such as the ones used in the Winograd schema (Levesque, 2011). A different approach is to use naturally occurring text, as used in recent machine reading tasks (see Section 9). One way to avoid the inherent biases people have when writing is to have them rate sentences from

naturally occurring text by parameters such as coherence (or, very differently, the level of surprise), rather than asking them to generate new text.

## 9 Related Work

**Writing style.** Writing style has been an active topic of research for decades. The models used to characterize style are often linear classifiers with style features such as character and word $n$-grams (Stamatatos, 2009; Koppel et al., 2009). Previous work has shown that different authors can be grouped by their writing style, according to factors such as age (Pennebaker and Stone, 2003; Argamon et al., 2003; Schler et al., 2006; Rosenthal and McKeown, 2011; Nguyen et al., 2011), gender (Argamon et al., 2003; Schler et al., 2006; Bamman et al., 2014), and native language (Koppel et al., 2005; Tsur and Rappoport, 2007; Bergsma et al., 2012). At the extreme case, each individual author adopts a unique writing style (Mosteller and Wallace, 1963; Pennebaker and King, 1999; Schwartz et al., 2013b).

The line of work that most resembles our work is the detection of deceptive text. Several researchers have used stylometric features to predict deception (Newman et al., 2003; Hancock et al., 2007; Ott et al., 2011; Feng et al., 2012). Some works even showed that gender affects a person's writing style when lying (Pérez-Rosas and Mihalcea, 2014a,b). In this work, we have shown that an even more subtle writing task—writing coherent and incoherent story endings—imposes different styles on the author.

**Machine reading.** The story cloze task, which is the focus of this paper, is part of a wide set of machine reading/comprehension challenges published in the last few years. These include datasets like bAbI (Weston et al., 2016), SNLI (Bowman et al., 2015), CNN/DailyMail (Hermann et al., 2015), LAMBADA (Paperno et al., 2016) and SQuAD (Rajpurkar et al., 2016). While these works have presented resources for researchers, it is often the case that these datasets suffer from methodological problems caused by applying noisy automatic tools to generate them (Chen et al., 2016). In this paper, we have pointed to another methodological challenge in designing machine reading tasks: different writing tasks used to generated the data affect writing style, confounding classification problems.

## 10 Conclusion

Different writing tasks assigned to an author result in different writing styles for that author. We experimented with the story cloze task, which introduces two interesting comparison points: the difference between writing a story on one's own and continuing someone else's story, and the difference between writing a coherent and an incoherent story ending. In both cases, a simple linear model reveals measurable differences in writing styles, which in turn allows our final model to achieve state-of-the-art results on the story cloze task.

The findings presented in this paper have cognitive implications, as they motivate further research on the effects that a writing prompt has on an author's mental state, and also her concrete response. They also provide valuable lessons for designing new NLP datasets.

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
