# Peer review of "The Effect of Different Writing Tasks on Linguistic Style: A Case Study of the ROC Story Cloze Task"

_ACL 2017 — decision unknown_

[Official Review · Reviewer 1 · rating 2 · confidence 4]
soundness 5 · originality 5 · clarity 2 · impact 3 · substance 4 · appropriateness 5 · meaningful comparison 3 · presentation format Poster

The paper analyzes the story endings (last sentence of a 5-sentence story) in
the corpus built for the story cloze task (Mostafazadeh et al. 2016), and
proposes a model based on character and word n-grams to classify story endings.
The paper also shows better performance on the story cloze task proper
(distinguishing between "right" and "wrong" endings) than prior work.

Whereas style analysis is an interesting area and you show better results than
prior work on the story cloze task, there are several issues with the paper.
First, how do you define "style"? Also, the paper needs to be restructured (for
instance, your section
"Results" actually mixes some results and new experiments) and clarified (see
below for questions/comments): right now, it is quite difficult for the reader
to follow what data is used for the different experiments, and what data the
discussion refers to.

(1) More details about the data used is necessary in order to assess the claim
that "subtle writing task [...] imposes different styles on the author" (lines
729-732). How many stories are you looking at, written by how many different
persons? And how many stories are there per person? From your description of
the post-analysis of coherence, only pairs of stories written by the same
person in which one was judged as "coherent" and the other one as "neutral" are
chosen. Can you confirm that this is the case? So perhaps your claim is
justified for your "Experiment 1". However my understanding is that in
experiment 2 where you compare "original" vs. "right" or "original" vs.
"wrong", we do not have the same writers. So I am not convinced lines 370-373
are correct.

(2) A lot in the paper is simply stated without any justifications. For
instance how are the "five frequent" POS and words chosen? Are they the most
frequent words/POS? (Also theses tables are puzzling: why two bars in the
legend for each category?). Why character *4*-grams? Did you tune that on the
development set? If these were not the most frequent features, but some that
you chose among frequent POS and words, you need to justify this choice and
especially link the choice to "style". How are these features reflecting
"style"?

(3) I don't understand how the section "Design of NLP tasks" connects to the
rest of the paper, and to your results. But perhaps this is because I am lost
in what "training" and "test" sets refer to here.

(4) It is difficult to understand how your model differs from previous work.
How do we reconcile lines 217-219 ("These results suggest that real
understanding of text is required in order to solve the task") with your
approach?

(5) The terminology of "right" and "wrong" endings is coming from Mostafazadeh
et al., but this is a very bad choice of terms. What exactly does a "right" or
"wrong" ending mean ("right" as in "coherent" or "right" as in "morally good")?
I took a quick look, but couldn't find the exact prompts given to the Turkers.
I think this needs to be clarified: as it is, the first paragraph of your
section "Story cloze task" (lines 159-177) is not understandable.

Other questions/comments:

Table 1. Why does the "original" story differ from the coherent and incoherent
one? From your description of the corpus, it seems that one Turker saw the
first 4 sentences of the original story and was then ask to write one sentence
ending the story in a "right" way (or did they ask to provide a "coherent"
ending?) and one sentence ending the story in a "wrong" way (or did they ask to
provide an "incoherent" ending)? I don't find the last sentence of the
"incoherent" story that incoherent... If the only shoes that Kathy finds great
are $300, I can see how Kathy doesn't like buying shoes ;-) This led me to
wonder how many Turkers judged the coherence of the story/ending and how
variable the judgements were. What criterion was used to judge a story coherent
or incoherent? Also does one Turker judge the coherence of both the "right" and
"wrong" endings, making it a relative judgement? Or was this an absolute
judgement? This would have huge implications on the ratings.

Lines 380-383: What does "We randomly sample 5 original sets" mean?

Line 398: "Virtually all sentences"? Can you quantify this?

Table 5: Could we see the weights of the features? 

Line 614: "compared to ending an existing task": the Turkers are not ending a
"task"

Line 684-686: "made sure each pair of endings was written by the same author"
-> this is true for the "right"/"wrong" pairs, but not for the "original"-"new"
pairs, according to your description.

Line 694: "shorter text spans": text about what? This is unclear.

Lines 873-875: where is this published?

[Official Review · Reviewer 2 · rating 2 · confidence 4]
soundness 5 · originality 5 · clarity 3 · impact 3 · substance 2 · appropriateness 4 · meaningful comparison 3 · presentation format Poster

- Strengths:
The paper has a promising topic (different writing styles in finishing a story)
that could appeal to Discourse and Pragmatics area participants.  

- Weaknesses:
The paper suffers from a convincing and thorough discussion on writing style
and implications of the experiments on discourse or pragmatics. 
(1) For example, regarding "style", the authors could have sought answers to
the following questions: what is the implication of starting an incoherent
end-of-story sentence with a proper noun (l. 582)? Is this a sign of topic
shift? What is the implication of ending a story coherently with a past tense
verb, etc. 
(2) It is not clear to me why studies on deceptive language are similar to
short or long answers in the current study. I would have liked to see a more
complete comparison here.
(3) The use of terms such as "cognitive load" (l. 134) and "mental states" (l.
671) appears somewhat vague. 
(4) There is insufficient discussion on the use of coordinators (line 275
onwards); the paper would benefit from a more thorough discussion of this issue
(e.g. what is the role of coordinators in these short stories and in discourse
in general? Does the use of coordinators differ in terms of the genre of the
story? How about the use of "no" coordinators?)  
(5) The authors do not seem to make it sufficiently clear who the target
readers of this research would be (e.g. language teachers? Crowd-sourcing
experiment designers? etc.) 

The paper needs revision in terms of organization
(there are repetitions throughout the text).  Also, the abbreviations in Table
5 and 6 are not clear to me. 

- General Discussion:
All in all, the paper would have to be revised particularly in terms of its
theoretical standpoint and implications to discourse and pragmatics.

=====

In their response to the reviewers' comments, the authors indicate their
willingness to update the paper and clarify the issues related to what they
have experimented with. However, I would have liked to see a stronger
commitment to incorporating the implications of this study to the Discourse and
Pragmatics area.

[Official Review · Reviewer 3 · rating 5 · confidence 4]
soundness 5 · originality 5 · clarity 5 · impact 3 · substance 4 · appropriateness 5 · meaningful comparison 3 · presentation format Oral Presentation

- Strengths:

*The paper is very well written
*It shows how stylometric analysis can help in reasoning-like text
classification
*The results have important implications for design on NLP datasets
*The results may have important implications for many text classification tasks

- Weaknesses:
*I see few weaknesses in this paper. The only true one is the absence of a
definition of style, which is a key concept in the paper

- General Discussion:
This paper describes two experiments that explore the relationship between
writing task and writing style. In particular, controlling for vocabulary and
topic, the authors show that features used in authorship attribution/style
analysis can go a long way towards distinguishing between 1) a natural ending
of a story 2) an ending added by a different author and 3) a purposefully
incoherent ending added by a different author.

This is a great and fun paper to read and it definitely merits being accepted.
The paper is lucidly written and clearly explains what was done and why. The
authors use well-known simple features and a simple classifier to prove a
non-obvious hypothesis. Intuitively, it is obvious that a writing task greatly
constraints style. However, proven in such a clear manner, in such a controlled
setting, the findings are impressive.

I particularly like Section 8 and the discussion about the implications on
design of NLP tasks. I think this will be an influential and very well cited
paper. Great work.  

The paper is a very good one as is. One minor suggestion I have is defining
what the authors mean by “style” early on. The authors seem to mean “a
set of low-level easily computable lexical and syntactic features”.  As is,
the usage is somewhat misleading for anyone outside of computational
stylometrics. 

The set of chosen stylistic features makes sense. However, were there no other
options? Were other features tried and they did not work? I think a short
discussion of the choice of features would be informative.